# Darkness within: The Internal Mechanism between Dark Triad and Malevolent Creativity

**DOI:** 10.3390/jintelligence10040119

**Published:** 2022-12-05

**Authors:** Zhenni Gao, Xinuo Qiao, Xiaobo Xu, Ning Hao

**Affiliations:** 1Institute of Brain and Psychological Sciences, Sichuan Normal University, Chengdu 610000, China; 2Shanghai Key Laboratory of Mental Health and Psychological Crisis Intervention, School of Psychology and Cognitive Science, East China Normal University, Shanghai 200062, China; 3Department of Psychology, Shanghai Normal University, Shanghai 200234, China

**Keywords:** malevolent creativity, Dark Triad, moral identity

## Abstract

The Dark Triad has been found to be associated with malevolent creativity (MC) in terms of trait level, and its underlying mechanism remains unclear. Based on the cognitive–affective processing system theory and the existing studies, the current study aimed to explore the internal mechanism between the Dark Triad and MC behavioral tendencies/performance. The results revealed that the Dark Triad is positively related to MC behavioral tendencies through trait aggression and general creativity behavioral tendencies. Regarding MC performance, the Dark Triad is positively related to the originality of malevolent ideas through MC behavioral tendencies, but this effect is only significant at low-to-medium levels of moral identity. In line with moral identity theory, a higher moral identity may prevent individuals from acting immorally due to their desire to maintain their moral image, which may further suppress malevolent idea generation. Therefore, cultivating moral identity may be an effective approach to weaken the Dark Triad–MC performance association.

## 1. Introduction

Creativity is the ability to produce novel and useful work ([42]; [45]). Traditionally, creativity has been viewed as a purely bright ability that improves people’s lives. As a common saying goes, where there is sunshine, there is also shade. Creativity also has a dark side, namely, malevolent creativity (MC), which refers to intentionally causing damage in novel ways ([9], [8]; [19]). For example, terrorists display malevolent creativity when they invent new bombs to more effectively massacre citizens.

Given that creativity is, by nature, unexpected and rule-violating, the products of MC can be unpredictable and extremely dangerous ([14]; [16]; [47]). Thus, the factors associated with MC have been explored, including aggression ([31]; [20]), approach motivation ([19]), and emotional state of anger ([5], [6]). Previous studies have also demonstrated a creativity–antisociality connection, namely, that creativity and antisocial behaviors can enhance each other ([15]; [16]). Researchers further moved in a ‘darker’ direction and focused on the Dark Triad, which is a personality type characterized by self-centeredness, manipulation, and callousness ([27]; [37]). Both creativity and the Dark Triad were found to benefit from lower inhibition ([32]; [36]; [43]; [48]), which allows individuals to connect remote concepts (i.e., concepts with loose associations) and then generate highly creative ideas ([32]; [36]; [43]). Moreover, individuals with lower inhibition are more sensitive to reward and find it harder to learn from punishment, which fosters antisocial traits ([11]). In line with these previous studies, other studies have found that the Dark Triad is positively correlated with general creativity ([26]; [28]; [29]; [44]). Additionally, the Dark Triad is positively related to violence, cyberbullying, and a lack of moral values ([34]; [35]). These findings imply that the Dark Triad may promote both general creativity and antisocial behaviors. In terms of generating creative and malevolent ideas, MC can also be enhanced by the Dark Triad. To date, studies have preliminarily shown that the Dark Triad is positively associated with MC ([25]; [30]; [46]). However, these studies were limited in terms of the trait level and did not clearly explain the specific underlying mechanism between the Dark Triad and MC. Therefore, the current study placed special emphasis on exploring the internal mechanism between the Dark Triad and MC behavioral tendencies (i.e., trait MC). Furthermore, the relationship between the Dark Triad and malevolent creative performance (MC performance) was examined.

The cognitive–affective processing system (CAPS) has proposed several types of mediating units (e.g., competencies and affective responses) that are activated by situational stimuli and further determine behaviors ([33]). In addition, these units interact with each other. Based on the CAPS, the current study explored which units may participate in the interaction between the Dark Triad and MC behavioral tendencies. MC involves generating creative ideas, while the Dark Triad has been found to be positively associated with general creativity ([26]; [28]; [29]; [44]). As defined, the core feature of MC is intentional damage, which may be associated with aggression ([9], [8]; [19]). Correspondingly, previous studies have shown that MC is positively correlated with aggression ([31]; [20]). In addition, the Dark Triad is related to increased violence ([35]). Based on these studies, we assumed that behavioral tendencies related to general creativity and aggression may act as mediators between the Dark Triad and MC behavioral tendencies.

According to the CAPS, the mediating units interact and produce different behavioral responses ([33]). Individuals with higher MC behavioral tendencies are prone to MC behaviors ([18]), which are evaluated by the originality and harmfulness of idea generation ([12]; [40]). The Dark Triad has been found to be positively related to MC behavioral tendencies ([25]; [46]); therefore, MC behavioral tendencies may be an underlying mediator between the Dark Triad and MC performance (i.e., originality and harmfulness). Moreover, moral identity theory proposes that individuals with higher moral identity have a strong motivation to maintain their moral image according to their moral standards ([3]; [39]). A recent study found that moral reasoning significantly moderates the relationship between MC behavioral tendencies and performance ([50]). A higher moral identity may prevent individuals from putting a malevolent creative idea into practice. Thus, moral identity may be a moderator of the mediation model involving the Dark Triad, MC behavioral tendencies, and MC performance. Additionally, previous studies have found gender differences in the Dark Triad ([26]), aggression ([23]), and MC ([38]). Moreover, the level of violence decreases with age ([24]). Thus, gender and age were included in the data analysis.

The current study aimed to answer the following question: How is the Dark Triad related to MC at the trait and behavior levels? Based on the above-mentioned studies, we hypothesized that (H1) behavioral tendencies of general creativity and aggression are significant mediators between the Dark Triad and MC behavioral tendencies and that (H2) MC behavioral tendencies mediate the relationship between the Dark Triad and MC performance (i.e., originality and harmfulness), while moral identity moderates the MC behavioral tendencies–performance path.

## 2. Materials and Methods

### 2.1. Participants

A total of 217 Chinese college participants (mean age = 21.51 ± 2.23 years; 166 females) were recruited in the study. The recruiting poster was sent to the open college WeChat group, and anyone interested in the experiment could sign up through a QR code. Three participants were removed due to incomplete data. None of participants had a history of mental or neurological illness. The prior power analysis revealed that the sufficient sample size was 166 to obtain reliable results (1-β = .95, α = .05, Effect size f = .15; [7]), which means that the current sample size met the criterion. Written informed consents were obtained from all participants before the experiment. The experimental procedure was approved by the University Committee on Human Research Protection of the East China Normal University (HR 319-2019; HR2-0197-2021).

### 2.2. Measures

#### 2.2.1. Malevolent Creativity Task (MCT)

The MCT was adapted from the realistic presented problem task (RPPT). It was used to measure MC performance by requiring participants to solve open-ended realistic problems in creative and malevolent ways ([12]). MCT contained 20 randomly sequenced trials, and participants were asked to generate a most novel solution for each problem (e.g., Hong is going to battle with an outstanding player in a tennis final, who is hard to defeat. Please think of a novel way for Hong to make the opponent ‘accidently’ injured before the final). The performances on the MCT were evaluated by originality and harmfulness scores. Four trained raters independently assessed the originality (1 = not original at all, 5 = very original) and harmfulness (1 = not harmful at all, 5 = very harmful) scores of each idea on a 5-point scale. The originality and harmfulness of each trial were calculated by averaging the rating scores from the four raters. Then, the final originality and harmfulness scores for each participant were obtained by averaging the scores of all trials. The inter-rater consistency values for originality (inter-rater correlation coefficient, ICC = .86) and harmfulness (ICC = .85) were satisfactory.

#### 2.2.2. Malevolent Creativity Behavior Scale (MCBS)

The MCBS was used to measure malevolent creative behavioral tendencies (MC behavioral tendencies, e.g., When I am treated unfairly, I will retaliate in a different way; [18]). It contains 13 items that are rated on a 5-point Likert scale (1 = never, 5 = always). The sum of all items was calculated to indicate the level of MC behavioral tendencies, which reflects the potential for malevolent creation. The internal consistency reliability was satisfactory (Cronbach’s α = .88).

#### 2.2.3. Runco Ideational Behavior Scale (RIBS)

The RIBS was used to measure general creative behavioral tendencies (e.g., I have some ideas for new inventions; [41]). It contains 19 items that are rated on a 5-point Likert scale (0 = never, 4 = just about every day). The sum of all items was calculated to indicate the level of general creativity behavioral tendencies, which reflects the potential for creation. The internal consistency reliability was satisfactory (Cronbach’s α = .93).

#### 2.2.4. Buss–Perry Aggression Questionnaire (BPAQ)

The BPAQ was used to measure aggression (e.g., When people disagree with me, I can’t resist arguing with them; [4]). It contains 29 items that are rated on a 5-point Likert scale (1 = strongly disagree, 5 = strongly agree). The sum of all items was calculated to indicate the level of aggression. The internal consistency reliability was satisfactory (Cronbach’s α = .87).

#### 2.2.5. Chinese Version of the Dirty Dozen (DD12)

The DD12 was used to measure Dark Triad personality traits (e.g., I lack remorse; [13]). It contains 12 items that are rated on a 7-point scale (1 = strongly disagree, 7 = strongly agree). The sum of all items was calculated to indicate the level of the composite Dark Triad. The internal consistency reliability was satisfactory (DD12: Cronbach’s α = .71).

#### 2.2.6. Moral Identity Measures (MIM)

The MIM was used to measure moral personality ([2]). Nine words referring to different good characteristics were listed (e.g., friendly), and participants were asked to complete 10 items based on how important these characteristics are to themselves (5-point Likert scale, 1 = strongly disagree, 5 = strongly agree; e.g., I strongly desire having these characteristics). The sum of all items was calculated to indicate the level of moral personality. The internal consistency reliability was satisfactory (Cronbach’s α = .75).

### 2.3. Procedure

Upon arrival, each participant was required to sit in front of a table and instructed to complete the demographic questions (i.e., gender and age), MCT, RIBS, BPAQ, DD12, and MIM (all scales were Chinese versions).

Before performing the MCT, participants were explicitly informed that all situations were fictitious in this experiment, and that malevolent ideas generated in this experiment were not related to their personality or level of morality. After completing the experiment, participants were asked whether they were all right (none of them claimed to feel unwell). Besides this, participants were also informed that psychological consultation was available and free for them if they needed it.

## 3. Results

### 3.1. Correlations between MC Performance and Other Variables

Pearson correlation was used to quantify the relations between MC performance (i.e., MCT originality and harmfulness) and other variables (see Table 1). The results revealed that MCT originality was positively correlated with MC behavioral tendencies (i.e., MCBS score; *r* = .30, *p* < .001), aggression (i.e., BPAQ score; *r* = .16, *p* = .018), and Dark Triad (i.e., DD12 score; *r* = .16, *p* = .022); MCT harmfulness was positively correlated with MC behavioral tendencies (*r* = .18, *p* = .006) and creativity behavioral tendencies (i.e., RIBS score; *r* = .21, *p* = .002). Moreover, the correlation coefficient between the Dark Triad and MC behavioral tendencies was significantly higher than that between the Dark Triad and creativity behavioral tendencies (z = 5.49, *p* < .001; [10]; [22]).

### 3.2. Regression Analysis on MC Performance

Before the subsequent analysis, all the variables were standardized into *Z* scores. Hierarchical regression analysis using gender (0 = female, 1 = male), age, and other scale scores as independent variables was performed on MCT originality. The results of Step 3 revealed that MC behavioral tendencies (β = .31, *t* = 3.25, *p* = .001) positively predicted MCT originality (see Table 2).

Hierarchical regression analysis using gender (0 = female, 1 = male), age, and other scale scores as independent variables was performed on MCT harmfulness. The results of Step 3 revealed that none of these variables significantly predicted MCT harmfulness after controlling for age and gender (see Table 3). The variance inflation factors (VIFs) ranged from 1.02 to 2.08, which meant that multicollinearity did not seriously affect these results ([17]; [49]).

### 3.3. Relations between Dark Triad and Malevolent Creativity Behavioral Tendencies

Parallel multiple mediation analysis using the Dark Triad as an independent variable and BPAQ and RIBS as mediators was performed on MC behavioral tendencies (see Figure 1). PROCESS 2.16.3 was used to examine the parallel multiple mediation effect ([21]). The results revealed that both the total effect (*b* = .57, *t* = 10.24, *p* < .001; Figure 1A) and direct effect (*b* = .40, *t* = 7.28, *p* < .001; Figure 1B) were significant; aggression (*b* = .10; 95% confidence interval, 95%CI = [.10, .26]) and creative behavioral tendencies (*b* = .08; 95%CI = [.04, .13]) significantly mediated the relation between the Dark Triad and MC behavioral tendencies. The VIF ranged from 1.08 to 1.26, which meant that multicollinearity did not seriously affect the result.

### 3.4. Relations between Dark Triad and Malevolent Creativity Behavioral Performance

A moderated mediation model using the Dark Triad as an independent variable, MC behavioral tendencies as a mediator, moral identity (i.e., MIM scores) as a moderator, and MCT originality as a dependent variable was established (See Figure 2). Model 14 of PROCESS 2.16.3 was used to examine whether the mediation effect of MC behavioral tendencies was moderated by moral identity ([21]). The results revealed that the total effect was significant (*b* = .16, *t* = 2.30, *p* = .022; Figure 2A), whereas the direct effect was insignificant (*b* = −.03, *t* = −.41, *p* = .684; Figure 2B); the moderated mediation effect was significant (*b* = −.09; 95%CI = [−.19, −.01]; Figure 2B); and moral identity significantly moderated the path between MC behavioral tendencies and MCT originality (*b* = −.16; 95%CI = [−.30, −.02]). Specifically, MC behavioral tendencies significantly mediated the relation between the Dark Triad and MCT originality when moral identity was one SD lower than the mean (*b* = .29; 95%CI = [.17, .43]) or equal to the mean (*b* = .19; 95%CI = [.11, .29]); the mediation effect of MC behavioral tendencies was insignificant when moral identity was one SD higher than the mean (*b* = .10; 95%CI = [−.02, .21]). The variance inflation factors (VIFs) ranged from 1.07 to 1.58, which meant that multicollinearity did not seriously affect these results.

## 4. Discussion

The current study investigated the internal mechanism between the Dark Triad and MC (i.e., MC behavioral tendencies and MC performance). We observed that MC behavioral tendencies were positively correlated with the Dark Triad, aggression, and general creative behavioral tendencies. Moreover, the correlation coefficient between the Dark Triad and MC behavioral tendencies was higher than that between the Dark Triad and general creative behavioral tendencies. The results further revealed that the Dark Triad was related to MC behavioral tendencies through both aggression and general creative behavioral tendencies. Regarding MC performance, only MC behavioral tendencies positively predicted MCT originality after controlling for age and gender. Mediation analyses revealed that MC behavioral tendencies significantly mediated the relationship between the Dark Triad and MCT originality; the mediation effect of MC behavioral tendencies was only significant in low-to-medium levels of moral identity.

The results showed that the Dark Triad was significantly correlated with general creativity and MC behavioral tendencies, which was consistent with the previous findings ([25]; [29]; [44]; [46]). Both the Dark Triad and creativity involve disinhibition, allowing individuals to be more antisocial and to connect remote ideas ([11]; [32]; [36]; [43]). Additionally, the results revealed that the positive correlation between the Dark Triad and MC behavioral tendencies was significantly stronger than that between the Dark Triad and general creative behavioral tendencies. MC is ‘eviler’ than general creativity due to its core feature of causing deliberate damage ([9], [8]; [19]), which may imply that MC is closer to an ‘evil trait’ such as the Dark Triad. Besides this, this result may suggest that individuals with higher Dark Triad not only tend to be more creative, but also may have an increased tendency to perform MC.

Parallel multiple mediation analysis found that the relationship between the Dark Triad and MC behavioral tendencies was mediated by both aggression and general creative behavioral tendencies. This result partly supports our hypothesis (i.e., H1). Researchers have demonstrated that individuals with higher aggression tend to generate more harmful ideas in general creativity tasks ([31]). A higher Dark Triad score is positively related to higher trait aggression, which may increase the tendency to generate harmful ideas. As a subtype of creativity, MC also pertains to finding creative solutions ([9], [8]; [19]). A previous study found that behavioral tendencies related to general creativity positively predicted MC behavioral tendencies ([19]). This result may therefore suggest that the Dark Triad stimulates general creativity and continues to foster the potential to design original evil plans. Taken together, the Dark Triad allows individuals to be more aggressive and creative so that it is easier to generate both malevolent and novel ideas, which further cultivates MC behavioral tendencies. Furthermore, weakening Dark Triad–aggression or Dark Triad–general creativity links may be useful to restrict the positive effect of the Dark Triad on MC.

At the behavioral level, MC behavioral tendencies positively predicted MCT originality (after the effects of age and gender were controlled for) and fully mediated the relationship between the Dark Triad and MCT originality. In addition, moral identity significantly moderated the path of MC behavioral tendencies–MCT originality in the mediation model. This supports our hypothesis (i.e., H2). The Dark Triad refers to self-centeredness, manipulation, and callousness ([27]; [37]). These ‘evil’ features are bonded to the potential for malevolent creation, which is a possible explanation for the positive correlation between the Dark Triad and MC behavioral tendencies ([25]; [30]; [46]). Thus, MC behavioral tendencies could be converted into actual MC performance ([18]), namely, positively connecting to the originality of malevolent creative ideas. However, the path of MC behavioral tendencies–MCT originality is only significant when moral identity is low or medium, which is in line with moral identity theory ([3]; [39]). A higher moral identity motivates individuals to act morally so as not to harm their moral image. In addition, research has found that individuals with higher levels of moral reasoning are less prone to putting their MC ideation into practice ([50]). Therefore, moral identity may weaken the positive relation between social aversive traits (i.e., the Dark Triad and MC behavioral tendencies) and the originality of malevolent ideas. Unexpectedly, this moderated mediation model was only suitable for MCT originality and not for MCT harmfulness (inconsistent with H2). This result may imply that MC behavioral tendencies (measured by the MCBS; [18]) are more closely associated with the capacity to generate original malevolent ideas. Harmfulness may be more closely related to other factors, such as dehumanization and high neuroticism ([1]).

In summary, the findings of the current study support CAPS and moral identity theory. Individuals with higher levels of Dark Triad personality traits tend to have higher aggression and general creativity behavioral tendencies, which then further cultivate their MC behavioral tendencies. At the behavioral level, MC behavioral tendencies may be closer to the originality than the harmfulness of malevolent ideation. The Dark Triad promotes MCT originality by fostering MC behavioral tendencies, but this mediation effect is only significant with low-to-medium moral identity. Based on the above-mentioned results, cultivating moral identity may be an effective way to prevent MC performance. Beyond the laboratory, MC performance in natural settings should be further investigated in the future. However, several limitations should be mentioned: (1) almost 76% of participants were female, so future studies should include more male participants; (2) only explicit aggression was explored, but implicit aggression could also be an effective source of malevolent creation; and (3) in addition to moral identity, moral reasoning and moral emotions should be further investigated in future studies.

## Figures and Tables

**Figure 1 jintelligence-10-00119-f001:**
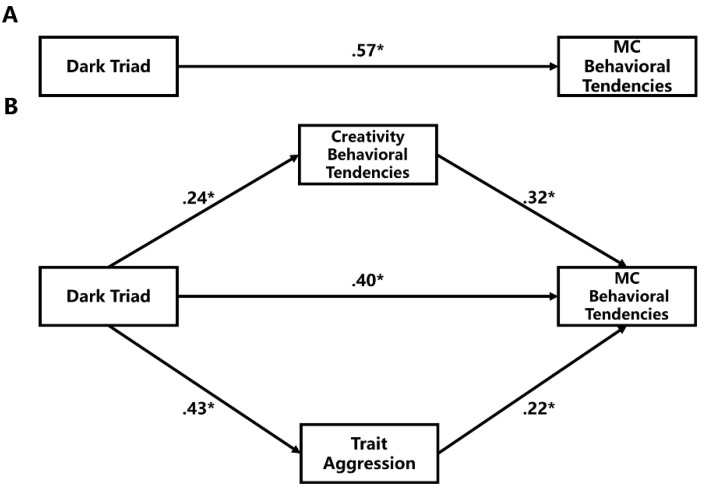
Results of parallel multiple mediation on MCBS: (**A**) the total effect of the Dark Triad on malevolent creativity potential; (**B**) the parallel multiple mediation model. Dark Triad = DD12 score; MC behavioral tendencies = MCBS score; Creativity behavioral tendencies = RIBS score; Trait aggression = BPAQ score; *: *p* < .05.

**Figure 2 jintelligence-10-00119-f002:**
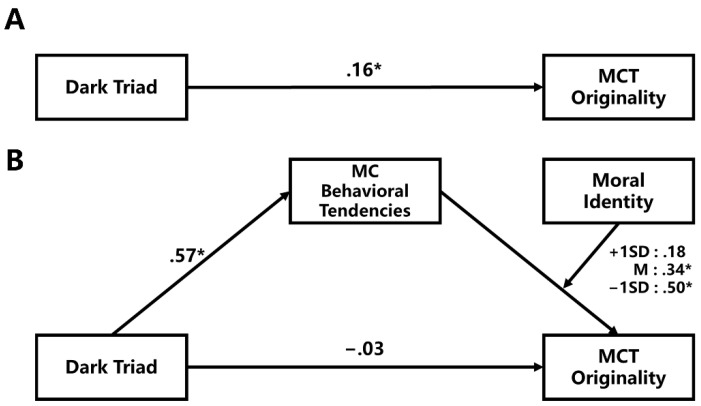
Results of moderated mediation analysis on MCT originality: (**A**) the total effect of the Dark Triad on MCT originality; (**B**) the moderated mediation model. Dark Triad = DD12 score; MC behavioral tendencies = MCBS score; Moral identity = MIM score; *: *p* < .05.

**Table 1 jintelligence-10-00119-t001:** Results of Pearson correlation analysis among all variables.

	2	3	4	5	6	7
1. MCT originality	.35 ***	.16 *	.30 ***	.10	.16 *	−.06
2. MCT harmfulness		.07	.18 **	.21 **	.05	−.03
3. Dark Triad			.57 ***	.23 ***	.43 ***	−.26 ***
4. MC behavioral tendencies				.47 ***	.47 ***	−.11
5. Creativity behavioral tendencies					.00	.08
6. Aggression						−.16 *
7. Moral identity						

Note: Dark Triad = DD12 score; MC behavioral tendencies = MCBS score; Creativity behavioral tendencies = RIBS score; Aggression = BPAQ score; Moral identity = MIM score. *: *p* < .05; **: *p* < .01; ***: *p* < .001.

**Table 2 jintelligence-10-00119-t002:** Results of hierarchical regression analysis using gender (0 = female, 1 = male), age, and the other scale scores as independent variables on MCT originality.

	Step 1		Step 2		Step 3	
	β	*t*	β	*t*	β	*t*
Gender	.15	2.16 *	.15 *	2.23	.09	1.28
Age	−.07	−1.08	−.05	−.71	−.03	−.40
Creativity Behavioral tendencies			.10	1.37	−.05	−.63
MCT Behavioral tendencies					.31 **	3.25
Dark Triad					−.04	−.46
Trait aggression					.04	.57
Δ *R*^2^	.02 *		.01		.07 *	
Δ *F*^2^	2.63		1.89		5.45	

Note: Dark Triad = DD12 score; MC behavioral tendencies = MCBS score; Creativity behavioral tendencies = RIBS score; Aggression = BPAQ score; Moral identity = MIM score. *: *p* < .05; **: *p* < .01.

**Table 3 jintelligence-10-00119-t003:** Results of hierarchical regression analysis using gender (0 = female, 1 = male), age, and the other scale scores as independent variables on MCT harmfulness.

	Step 1		Step 2		Step 3	
	β	*t*	β	*t*	β	*t*
Gender	.01	.11	.02	.23	−.01	−.20
Age	−.24	−3.58 ***	−.20 **	−2.94	−.19 **	−2.82
Creativity Behavioral tendencies			.16 *	2.29	.10	1.35
MCT Behavioral tendencies					.15	1.55
Dark Triad					−.03	−.39
Trait aggression					−.05	−.58
Δ *R*^2^	.06		.02		.01	
Δ *F*^2^	6.48 **		5.25 *		.83	

Note: Dark Triad = DD12 score; MC behavioral tendencies = MCBS score; Creativity behavioral tendencies = RIBS score; Aggression = BPAQ score; Moral identity = MIM score. *: *p* < .05; **: *p* < .01; ***: *p* < .001.

## Data Availability

The data presented in this study are available on request from the corresponding author. The data are not publicly available due to [the participant privacy and the data is only to be made available via a request with a formal data sharing agreement and the approval from the requesting researcher’s local ethics committee].

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
