# Peer review of "Darkness within: The Internal Mechanism between Dark Triad and Malevolent Creativity"

_jintelligence, 2022, doi:10.3390/jintelligence10040119_

Round 1

Reviewer 1 Report

Dear author(s),

I hope you are doing well, and thank you for your submission. I appreciate the time that you put into this manuscript. Please see the following suggestions to increase the quality of this manuscript:

 General

·          Please proofread for some format issue and make sure you are using a consistent style throughout the manuscript.

Introduction

·         Line 52, 55 and 66, the abbreviation for “cognitive-affective processing system” should be CAPS. (Please check this throughout the whole manuscript.)

·         Revise your H1 and remove “may be”.

Materials and Methods

·         Sampling strategy is not clear. How did you recruit these participants?

·         Add more demographic information. Are they college students? Ethnicity? Give readers an appropriate picture of the participants/context.

·         Line 146, you mentioned “All the items were presented in Chines.” Is there Chinese version of all of the measures you mentioned? If so, please clarify. If not, what was the procedure. How did you translate? Please elaborate on the procedure in more details.

Results and Data Analyses

·         Be consistence in reporting statistics and statistical-analysis results throughout the results section.

·         Line 171-172, add a citation to support.

·         Please justify why you just included age and gender for your regression analysis.

Discussion

·         Line 244-245, you mentioned that “Combined with the prosocial psychopath model, this result may suggest that MC shares a more common mental process with the Dark Triad than with general creativity.” What would be potential application/implication of this?

·         Line 255-256, you mentioned that “This suggests that the Dark Triad stimulated general creativity and continued to foster the potential to design creative evil plans.” What would be the potential application?

·          Line 259-280, make sure to remove cause-and-effect language, as you haven’t conducted a true experimental design.

Best wishes,
Your peer

Reviewer 2 Report

I believe this might be a publishable paper in JoI but I am really concerned about ethical issues. The idea and the data are interesting and with some minor English language editing, the paper will be easy to read.

I have some concerns, as follows.

1. I do not read papers for their moral stance. But I found the idea of a "prosocial psychopath" model to be morally reprehensible and psychologically flawed.  It is a serious misnomer.  If traits associated with psychopathy can lead to creativity, that is one thing. That is a scientific association.  But to call that "prosocial" when you say right below it leads to "antisocial" traits seems contradictory.  Creativity certainly requires defiance of norms.  But defiance of norms is not in itself prosocial (or necessarily antisocial), nor is creativity. Creativity can be highly antisocial.  Look at the drone attacks in Ukraine--creative?  Maybe?  Prosocial? No way.  It may be someone else's theory, but to use it blindly is a mistake. There is nothing prosocial about antisocial behavior.  It is self-contradictory. The fact that psychopathic traits can lead to creativity does not make those traits prosocial.

2. I have an ethical concern about the MCBS.  Essentially, the instrument seems to invite participants to think in malevolent and destructive ways. What assurance do you have that your inviting and encouraging participants to think of harmful acts (arranging supposed accidents), for example, won't be taken as a kind of permission on the part of the psychological establishment to think and act this way?  What if you give participants ideas about acting malevolently that they then believe they have permission to act upon?  You do not give them a choice to act positively.  You actually more or less force them to think in malevolent ways, so you may be getting experimenter effects of an antisocial kind.  What kind of message does this send about psychologists?  That we want to plant the seeds for antisocial behavior?

3. Was this study approved by an IRB?  Did they have nothing to say about this issue?

4.  It is not clear to me exactly what you are studying.  The MCBS does not seem to me to measure actual malevolent creativity but rather the participant's ability to come up with malevolent ideas if encouraged to do so by an authority figure, namely, the researcher.  It may be heavily grounded in experimenter effects.  It may establish a permission to think in this way that participants previously did not feel they have.

5. We live in a time in which malevolent creativity is becoming more and more pervasive in the world.  I think we should be very careful about doing anything that may encourage it.

6.  I realize that to do research on malevolent creativity, one needs scales.  I just do not see how this study took proper precautions to ensure that it did not have negative effects on the participants. Did it? I cannot tell for sure.

7. The results were interesting and the correlational data make sense.

Round 2

Reviewer 1 Report

Dear author(s), 

Thank you for submitting your revision and addressing my suggestions. 

Best regards, 

Your peer

Reviewer 2 Report

This version is improved.

In your reply, you say:

"Before the MCT, we informed participants explicitly that all task situations were unreal and the malevolent ideas they generated were not related to their personality or the level of morality. After all scales and tasks finished, we asked participants whether they were all right and none of them claimed to feel unwell. Besides, we told participants that the psychological consultation was available and free for them if they needed it."
This should be stated in the Procedure section so that readers know that care was taken regarding ethics of the research. I am glad that the IRB approval was added to the Method section.  Thank you for taking out the prosocial psychopath model, which does not make sense.

The paper needs editing for English-language quality.

our return visit, none of them involve in psychological consultation or felt distressful due
to the participation in our experiment.
(3
